# Mucopolysaccharidosis-Plus Syndrome: Report on a Polish Patient with a Novel *VPS33A* Variant with Comparison with Other Described Patients

**DOI:** 10.3390/ijms231911424

**Published:** 2022-09-28

**Authors:** Patryk Lipiński, Krzysztof Szczałuba, Piotr Buda, Ekaterina Y. Zakharova, Galina Baydakova, Agnieszka Ługowska, Agnieszka Różdzyńska-Świątkowska, Zuzanna Cyske, Grzegorz Węgrzyn, Agnieszka Pollak, Rafał Płoski, Anna Tylki-Szymańska

**Affiliations:** 1Department of Pediatrics, Nutrition and Metabolic Diseases, The Children’s Memorial Health Institute, Al. Dzieci Polskich 20, 04-730 Warsaw, Poland; 2Department of Medical Genetics, Medical University of Warsaw, 02-106 Warsaw, Poland; 3Research Centre for Medical Genetics, 115522 Moscow, Russia; 4Department of Genetics, Institute of Psychiatry and Neurology, 02-957 Warsaw, Poland; 5Anthropology Laboratory, The Children’s Memorial Health Institute, 04-736 Warsaw, Poland; 6Department of Molecular Biology, Faculty of Biology, University of Gdańsk, 80-309 Gdańsk, Poland

**Keywords:** mucopolysaccharidosis, mucopolysaccharidosis-plus syndrome, *VPS33A*, lysosomal storage disease

## Abstract

Eleven patients from Yakutia with a new lysosomal disease assumed then as mucopolysaccharidosis-plus syndrome (MPS-PS) were reported by Gurinova et al. in 2014. Up to now, a total number of 39 patients have been reported; in all of them, the c.1492C>T (p.Arg498Trp) variant of the *VPS33A* gene was detected. Here, we describe the first Polish MPS-PS patient with a novel homozygous c.599G>C (p.Arg200Pro) *VPS33A* variant presenting over 12 years of follow-up with some novel clinical features, including fetal ascites (resolved spontaneously), recurrent joint effusion and peripheral edemas, normal growth, and visceral obesity. Functional analyses revealed a slight presence of chondroitin sulphate (only) in urine glycosaminoglycan electrophoresis, presence of sialooligosaccharides in urine by thin-layer chromatography, and normal results of lysosomal enzymes activity and lysosphingolipids concentration in dried blood spot. The comparison with other MPS-PS described cases was also provided. The presented description of the natural history of MPS-PS in our patient may broaden the spectrum of phenotypes in this disease.

## 1. Introduction

Mucopolysaccharidoses (MPSs) are a group of lysosomal storage diseases (LSDs) caused by the deficiency of specific enzymes involved in catabolism of glycosaminoglycans (GAGs) [1,2,3]. The deficiency of each enzyme leads to the accumulation of undegraded GAGs, including heparan sulfate (HS), dermatan sulfate (DS), keratan sulfate (KS), chondroitin sulfate (CS), or hyaluronan and secondary disturbances of cellular homeostasis [4,5]. The clinical features of MPSs differ depending on the specifying enzyme deficiency and accumulated type of GAGs; however, the most specific symptoms include a short stature (with the most severe growth impairment in MPS IVA), skeletal symptoms (*dysostosis multiplex*, joint stiffness, but hypermobility and unique skeletal abnormalities in MPS IVA), neurodegeneration (progressive cognitive decline, especially in MPS III but also MPS IH, neuronopathic MPS II), coarse facial features, and cardiac (valvular) disease (MPS I, II, VI). An increased urinary GAGs excretion supports the clinical suspicion of MPSs, and subsequent enzymatic and molecular analyses are essential to establish a diagnosis of a specific type of MPS [4].

Aside from lysosomal enzyme deficiencies, defects of integral lysosomal membrane proteins and other proteins acting as transporters or activators, as well as defects in the vesicular cell trafficking related to lysosomes, are described as causative of LSD [6].

In 2017, a new disease assumed as mucopolysaccharidosis-plus syndrome (MPS-PS) was included in the Online Mendelian Inheritance in Man^®^ (OMIM^®^) database (Warsaw, Poland) [7]. The name of the disease stems from the presence of only some clinical or biochemical features resembling MPS (including dysostosis multiplex in some cases, thickened facial features, valvular disease, psychomotor retardation, slightly increased urinary GAGs excretion), and additional signs and symptoms. Up to now, a total number of 38 patients have been reported [8,9,10,11,12,13,14]. In all of them, the disease was caused by the NM_022916.6:c.1492C>T (p.Arg498Trp) variant of the *VPS33A* gene, coding for the vacuolar protein sorting-associated protein 33A. When this paper was being reviewed, a preprint reporting an MPS-PS patient with the homozygous c.599G>C (p.Arg200Pro) variant of *VPS33A* appeared (https://www.medrxiv.org/content/10.1101/2022.08.27.22279208v1 (accessed on 21 September 2022)).

Here, we describe the first Polish MPS-PS patient affected with a novel homozygous *VPS33A* variant presenting with some novel clinical features as well as a slower progression rate with longitudinal follow-up. We also provide a comparison with other MPS-PS cases reported so far.

## 2. Case Report

The patient was born from the first pregnancy of nonconsanguineous (but probably related, due to a multigenerational close proximity of both families from Mazovia) Polish parents (Figure 1A). A prenatal period was complicated by fetal ascites noted on ultrasound in the 11th and 24th weeks of gestation. The pregnancy was terminated by cesarean section in the 40th week of gestation due to lack of delivery progress. The child was macrosomic and large for gestational age (birth weight 4830 g (>97th percentile (pc)), length 60 cm (>97th pc]) head circumference 38 cm (>97th pc)). It should be taken into account that the parents are tall with a high body weight, so the family background may have had an impact on the parameters of the child. An abdominal ultrasound in the newborn did not indicate any abnormalities.

At the age of 2.5 months, the patient was diagnosed with pneumonia. Laboratory results at that time revealed a subclinical hypothyroidism. Echocardiography (performed due to heart murmur) showed a mild stenosis of the mitral valve. The second bout of pneumonia at 5 months of age was accompanied by urinary tract infection; bilateral vesicoureteral reflux was diagnosed. During the third episode of pneumonia, at 12 months of age, the liver enlargement was noted (liver was palpable 5 cm below the costal arch) with elevated serum transaminases (AST 179 U/L and ALT 243 U/L) but with normal liver synthetic function.

From an early-infantile period, a delayed psychomotor development with autistic signs (similar to in the other family members) were observed; an independent sitting was observed from 13 months of age while walking was observed from 28 months of age. Anthropometric measurements in the first years of life, including body mass and length/height, were both above the 97th pc (see Figure 2).

At the age of 7.5 years, the patient was hospitalized due to pneumonia and sepsis accompanied by facial and lower limb edemas. At that time, hypogammaglobulinemia and glomerular proteinuria were noted. An ultrasound examination revealed the hip and knee joint effusion. Juvenile idiopathic arthritis was suspected, and treatment with hydroxychloroquine was implemented. One year later, the patient presented with aggravating lower limb edemas and secondary hip joint stiffness. The hip joint effusion was still observed by ultrasound, and treatment with steroids was implemented. During several months of treatment, a gradual reduction of edemas was observed. At the age of 10 years, the patient had another episode of pneumonia associated with pronounced facial, palpebral, and lower limb edemas. At that time, magnetic resonance examination of the hip joints revealed an asymmetric sclerotic remodeling of the deformed heads of both femurs, synovial hypertrophy, and accumulation of fluid. Diagnosis of juvenile idiopathic arthritis was questioned.

At the age of 12 years (March 2022), the patient was hospitalized in our department due to progressive generalized massive edemas involving the face, trunk, and limbs. A gradual regression of motor abilities was observed. Body composition analysis performed using the BIA revealed a body mass of 67.2 kg (weight gain by 6.5 kg over 10 months; see Figure 2 and Figure 3) with an excess body fat (12.3 kg) as well as visceral fat area of 130 cm^2^ (normal results < 100 cm^2^) and increased total body water (TBW) volume (33.7 L, with increase by 8.4 L over 10 months; reference range 25.2–30.8 L). Furosemide and spironolactone treatment was implemented with a moderate effect; after 3 months of treatment (June 2022), the body weight decreased by 6 kg and TBW volume dropped by 5.6 L (see Figure 3). Laboratory results showed a persisted hypogammaglobulinemia, slight proteinuria (0.25–0.28 g/24 h, normal results < 0.15 g/24 h) with normal serum albumin concentrations, normal liver function tests, and increased uric acid levels (6.8–7.4 mg/dL, reference values 2.6–5.9 mg/dL). Echocardiography revealed no progression of mitral valve stenosis. Ultrasound examination revealed no effusion in the hip and knee joints’ cavities.

There was a moderate increase in urinary GAGs excretion of 20.9 mg/mmol creatinine (reference values < 9.9 mg/mmol creatinine). GAGs electrophoresis showed the presence of chondroitin sulphate; see Figure 4 (which can also be excreted in small amounts in healthy individuals). Thin-layer chromatography showed the presence of sialooligosaccharides in urine; see Figure 5.

Normal activity of lysosomal enzymes involved in GAGs catabolism was demonstrated. Subsequently, a quantification of lysosomal enzymes activities and lysosphingolipids concentrations in DBS were performed revealing normal results (see Table 1).

Due to an unknown etiology of presented features, whole-exome sequencing (WES) was commenced. Based on the WES data, the homozygous variant in *VPS33A* gene (NM_022916.6:c.599G>C) were highlighted for further verification. ADS confirmed presence of the homozygous c.599G>C variant within the proband’s DNA and revealed the presence of this variant in heterozygous form in both parents (see Figure 1). The c.599G>C variant in the *VPS33A* gene was not described heretofore (HGMD Professional 2021.4, www.hgmd.cf.ac.uk (accessed on 19 August 2022)). The population frequency of the variant c.599G>C in the *VPS33A* gene was 0.00000398 in gnomAD Exomes database (Version: 2.1.1). In silico analysis of the pathogenicity using MetaLR, MetaSVM, MetaRNN, and REVEL algorithms indicated that the *VPS33A* c.599G>C variant is damaging/pathogenic. The American College of Medical Genetics and Genomics classification (ACMG) evaluates it as pathogenic (PM2, PP3). The *VPS33A* c.599G>C was registered once in ClinVar database (Version: 06-Jul-2022) and classified as a variant with uncertain significance (accession no. RCV002013360.2).

Finally, based on all the given above findings, the mucopolysaccharidosis-plus syndrome (MPS-PS) was diagnosed.

## 3. Comparison with Other Reported MPS-PS Cases

Up to now, 39 patients with MPS-PS have been described [8,9,10,11,12,13,14,15]. Notably, a detailed clinical presentation and follow-up was not available in some of the reported cases, especially in the first report of 11 Yakut children authored by Gurinova et al. [8]. A comparison between clinical features of our patient and all of the up-to-date reported individuals is summarized in Table 2.

## 4. Discussion

In this study, we described the first Polish MPS-PS patient presenting with a novel homozygous *VPS33A* variant. So far, all the reported MPS-PS patients were found among the Yakut population (Russia) and two patients from Turkey; it is to be noted that Yakuts and Turks belong to the Turkic ethnic group. All of them were homozygotes for c.1492C>T, p.(Arg251Glu) variant in the *VPS33A* gene. This phenomenon is most likely determined by a founder effect [16]. However, Faraguna et al. recently reported an additional MPS-PS patient of Moroccan origin affected with this variant in homozygosity [13]. When this paper was being reviewed, a preprint reporting an MPS-PS patient with the homozygous c.599G>C (p.Arg200Pro) variant of *VPS33A* appeared, which corroborated conclusions presented in our work [15].

In contrast to all reported patients, the clinical phenotype of our patient seems to be attenuated with a slower progression rate and the longest life span (12 years at the last follow-up). Almost all reported patients died by 2 years of age due to respiratory/cardio-respiratory failure; see Table 2.

The characteristic and previously nonreported features of our patient comprised the extracellular fluid excess presenting as fetal ascites (which resolved spontaneously), joint effusion, and peripheral edemas. The pathophysiology of this phenomenon is not fully understood.

Similar to other MPS-PS patients reported in the literature, our patient presented with congenital heart defect (no progression of valve disease was observed), subclinical hypothyroidism, and renal phenotype (only subtle proteinuria). We also found a decreased serum IgG concentration in our patient (similar to the patient reported by Pavlova et al. [11]). A gradual regression of motor abilities was observed during follow-up.

Sofronova et al. recently reported on progressive hematopoietic abnormalities noted in MPS-PS patients, especially in the red blood cells and platelets; however, we did not observe any of them [14]. They also observed that most patients experienced elevated levels of uric acid in the blood as a manifestation of kidney failure. However, except the elevated serum uric acid level, our patient had normal kidney function. The novel feature of our patient comprised the visceral obesity. Due to its presence, we could speculate about an increased uric acid level as a putative biomarker of pre-emptive metabolic syndrome. The other hypothesis comprises an increased cell turnover in MPS-PS.

Growth retardation, as well as poor weight gain, were also observed in several patients reported by Sofronova et al. [14]. Contrary to that, our patient had normal growth (see Figure 2) and was overweight (visceral obesity).

MPS-PS pathophysiology is not known and remains to be elucidated. In all the reported patients (as well as in our patient), lysosomal enzyme activities were within normal ranges. MPS-PS patients showed an increased urinary GAGs excretion (probably secondary) while GAGs electrophoresis revealed various results. Pavlova et al. found an increased HS and DS in three patients while one of them also showed a trace of KS [11]. Kondo et al. found extremely high levels of HS in patients’ plasma (when compared with MPS type II patient and the normal reference range) [10]. HS was also accumulated in patient-derived fibroblasts and VPS33A-depleted cells. Contrary to these reports, in our patient we observed a slight presence of CS (only) in urine GAGs electrophoresis. We would like to highlight that there is no correlation between clinical and biochemical (including urinary GAGs excretion) features of MPS-PS. Furthermore, HS has long been associated with neurological impairment in MPS. The presence of familial autistic features with the absence of HS accumulation adds weight to GAGs accumulation being a consequence of disease and not a cause.

Pavlova et al. revealed a pathological excess of sialylated conjugates in urine which was also found in our patient [11].

In the patient reported by Pavlova et al., mass spectrometry analysis showed high levels of β-D-galactosylsphingosine in skin fibroblasts [11]. Unlike that report, our patient presented normal results of lysosphingolipids in serum and DBS.

In addition, an MPS-PS-like disease was found in two siblings from a Turkish–Arabic family which was described as resembling MPS-PS syndrome caused by a homozygous c.540G>T (p.Trp180Cys) variant in the *VPS16* gene [17]. Another couple of patients, with symptoms resembling MPS-PS, harboring a pathogenic splice variant of *VPS16,* were reported recently [18]. VPS16 next to VPS33A, VPS11, VPS18, VPS39, and VPS41 is included in the HOPS (homotypic fusion and vacuole protein sorting) tethering complex which mediates docking interactions among late endosomes, lysosomes, and autophagosomes [17,18]. The disorganization of the HOPS complex causes the impairment of the fusion of lysosomes with endosomes or autophagosomes and, therefore, it is likely that HOPS dysfunction might be primary biochemical cause of MPS-PS and MPS-PS-like disorders. However, Kondo et al. reported that the c.1492C > T p.(Arg498Trp) variant of the *VPS33A* gene did not alter the autophagy process [10]. Thus, further research is necessary to determine the molecular mechanism of MPS-PS.

## 5. Material and Methods

### 5.1. Clinical Case

This is a retrospective analysis of clinical and biochemical data obtained from medical records of an MPS-PS patient diagnosed and followed up at the Children’s Memorial Health Institute (CMHI), Warsaw, Poland.

Ethical approval was obtained from the Children’s Memorial Health Institute Bioethical Committee, Nr 23/KBE/2020, Warsaw, Poland.

### 5.2. Lysosomal Enzymes and Lysosphingolipids in Dried Blood Spot (DBS)

The internal standard (IS), substrates, and assay buffers of panel 1 (GALC, GAA, GLA, ABG, ASM, and IDUA) and panel 2 (I2S, NAGLU, GALNS, ARSB, GLB1, GUSB, and TPP1) were commercially purchased from PerkinElmer, Inc. (Waltham, MA, USA).

The protocol of the multiplex enzyme assay is as follows. All of the assays were carried out with a 3.2 mm punch in 30 μL of assay cocktail in a polypropylene 96-well plate and incubated at 37 °C for 18 h. To terminate this enzyme reaction, a mixture of methanol/ethyl acetate (50/50, 100 μL) was added. Solvent extraction was carried out by adding 400 µL of ethyl acetate and 300 µL of deionized water (for panel 1) and 400 µL of ethyl acetate and 300 µL of 0.5M NaCl (for panel 2), followed by centrifugation for 5 min at 4000 rpm. A total of 300 µL of the organic layer was transferred to a new 96-well plate and dried under a stream of nitrogen at 40 °C. The dried precipitate was dissolved in 100 µL solvent mixture (acetonitrile: deionized water (80:20) containing 0.2% formic acid). The enzyme activity was calculated by the following formula: A = ((P/IS) × CIS)/(3.1 × ti), where A—enzyme activity expressed as micromoles of product per liter of whole blood per hour (µM/l/h), P—product peak area produced during the enzymatic reaction, IS—peak area of internal standard, CIS—the concentration of the internal standard in μM, ti—incubation time, and 3.1—volume of sample in µL.

Samples were measured using an LC-30 Nexera System (Shimadzu Corporation, Kyoto, Japan) and a tandem mass spectrometer QTrap 4500 (ABSciex, Framingham, MA, USA) equipped with a positive electrospray ionization by selected ion monitoring mode (Multiple Reaction Monitoring, MRM, New York, NY, USA); LC-MS/MS conditions are described in the Appendix A.

### 5.3. Oligosaccharides and Sialooligosaccharides in Urine

The material for the study was urine samples collected after the night without the addition of preservatives. Urine samples were stored at −20 °C until analysis.

Thin-layer chromatography (TLC) of oligosaccharides and sialooligosaccharides was performed according to the methods described by Blom et al., Humbel and Collart, and Svennerholm with our own modification, which was the introduction of a preliminary step of desalting the urine samples on mini-columns with anionite AG 1X8, 200–400 mesh (Bio Rad, Hercules, CA, USA) and cationite Dowex, 50X8-200, 100–200 mesh (Sigma-Aldrich, St. Louis, MI, USA) [19,20,21,22].

### 5.4. Bioelectric Impedance Analysis

Body composition analysis by the electrical bioimpedance method (BIA) using an InBody S10 (Biospace, Urbandale, Iowa) device was conducted. This device enables comprehensive body water analysis for monitoring body water status and nutritional status. InBody S10 uses multiple frequencies: 1 kHz, 5 kHz, 50 kHz, 250 kHz, 500 kHz, and 1000 kHz.

The bioelectrical method assesses the tissue components of the body, involving a measurement of the electrical resistance in the human body. The resistance varies depending on the composition of the subject’s body. The assessment of body composition and the proportion between the fatty and muscle tissues is obtained after entering data on age and gender; the device measures body height and weight and performs measurements of resistance and reactance. The test provides information on body composition: body water (L); the amount of protein (kg); the amount of mineral substances (kg); fatty tissue mass (kg); % of body fat in whole body weight; skeletal muscle mass (kg); and the distribution of the fatty and muscle tissues in the various segments of the body. The measurement was performed on a patient, according to the instructions of the measurement. The patient assumed a sitting position with touch electrodes clipped onto the patient’s fingers and ankles. The results were shown as body water analysis: total body water (sum of intracellular water and extracellular water), segmental water, which evaluates whether the amount of body water was adequately distributed throughout the body, and ECW ratio (the ratio of extracellular water to total body water, which is an indicator of body water balance) [23].

### 5.5. Genetic Analysis

DNA from the proband and her parents was retrieved from peripheral blood and extracted with a standard protocol. Library preparation for the whole-exome sequencing (WES) was performed on the proband’s DNA sample with Twist Human Core Exome spiked-in with Twist mtDNA Panel, Twist RefSeq Panel, and Custom Panel covering variants located in noncoding regions that have been linked to clinical phenotypes according to the ClinVar database (Twist Bioscience, San Francisco, CA, USA). Enriched library was paired-end sequenced (2 × 100 bp) on NovaSeq 6000 (Illumina, San Diego, CA, USA) to obtain 77,960,870 reads resulting in mean depth of 76.52× (98.6% of target bases were covered at a minimum of 20×, whereas 99.1% had coverage of minimum 10×). Bioinformatic analysis of raw WES data and variants prioritization were performed as previously described [24]. Reads were aligned to the hg38 reference genome sequence and visualized by Integrative Genomic Viewer. Presence of the selected variant was confirmed and established by amplicon deep sequencing (ADS) in the proband and her parents.

### 5.6. Comparison with Other Reported MPS-PS Cases

The database PubMed (MEDLINE) was searched for relevant studies. The following keyword-based strategy was used: (“Mucopolysaccharidosis-Plus Syndrome” OR “VPS33A disease”). Only original studies that contained sufficient data were included. The following data were extracted and then compared: number of reported patients, ethnicity, consanguinity, renal phenotype, hematological phenotype, cardiological phenotype, immunological phenotype, and follow-up.

## 6. Conclusions

We described an additional MPS-PS patient presenting with not-previously-reported recurrent edemas usually triggered by infections, normal growth with visceral obesity, and gradual regression of motor abilities. These features could be related to the patient’s genotype (novel homozygous *VPS33A* variant) but also could be observed over long-term follow-up.

The presented description of the natural history of MPS-PS in our patient may broaden the spectrum of phenotypes in this disease.

## Figures and Tables

**Figure 1 ijms-23-11424-f001:**
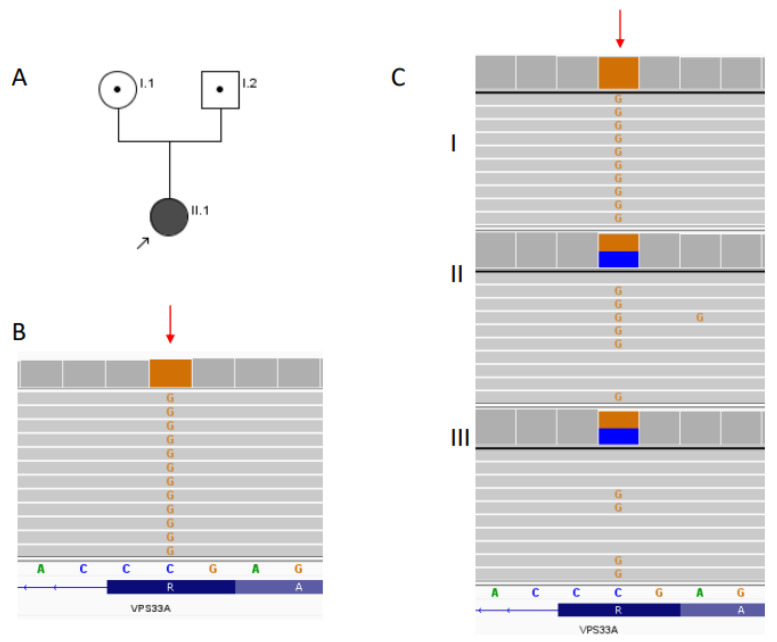
Pedigree of the studied family (**A**), results of WES in the proband (**B**), results of deep amplicon sequencing in the proband and her parents (**C**): **I**—proband, **II**—mother, **III**—father. Red arrows indicate c.599 position in *VPS33A* gene.

**Figure 2 ijms-23-11424-f002:**
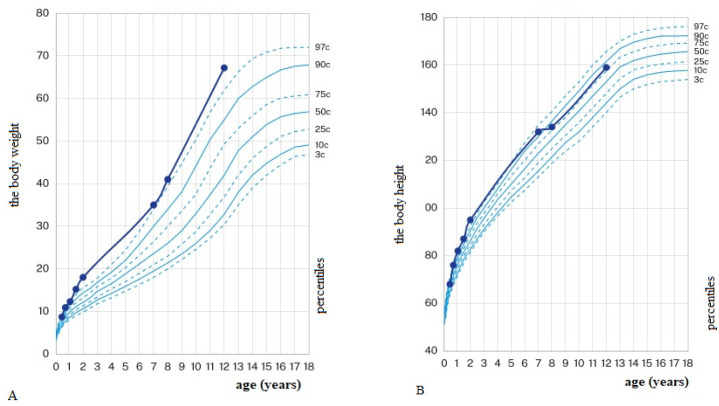
Percentile charts for the body weight (**A**) and the body height (**B**).

**Figure 3 ijms-23-11424-f003:**
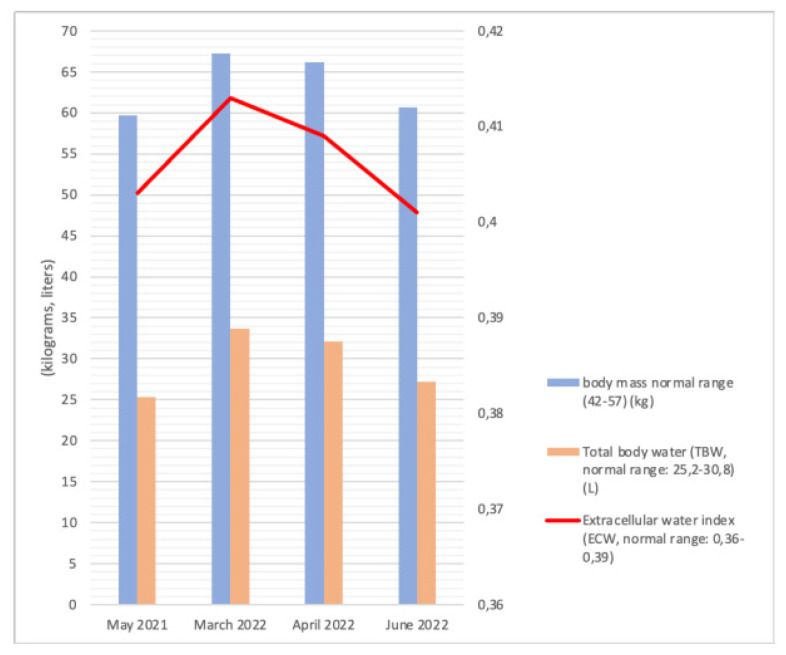
Relationship between body weight, total body water, and the extracellular water ratio.

**Figure 4 ijms-23-11424-f004:**
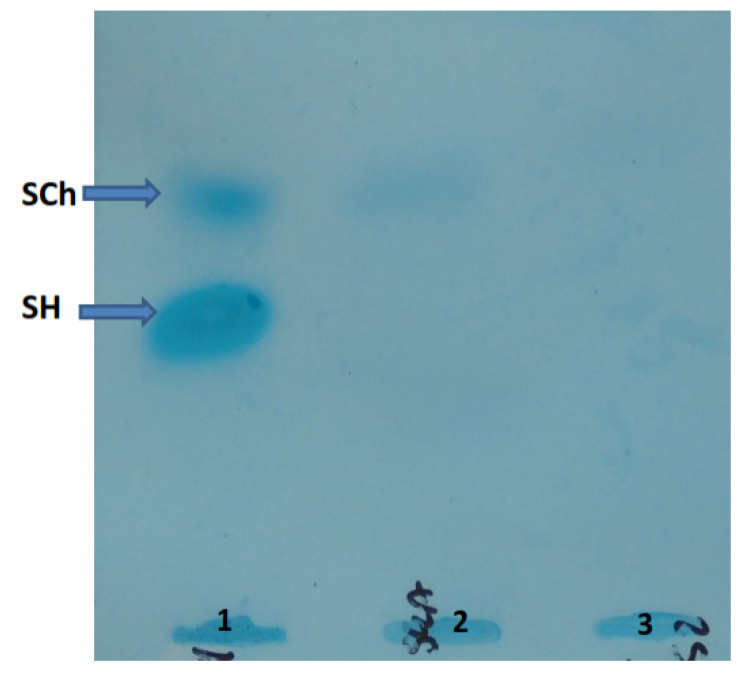
Picture of glycosaminoglycans (GAGs) after electrophoresis on cellulose acetate strips. GAGs were isolated from urine sediment, electrophoresed on cellulose acetate strips, and visualized with alcian blue staining. Path 1—patient with MPS III, Path 2—in patient described here; only a slight band of SCh is visible, Path 3—healthy donor. SCh—chondroitin sulphate, SH—heparan sulphate.

**Figure 5 ijms-23-11424-f005:**
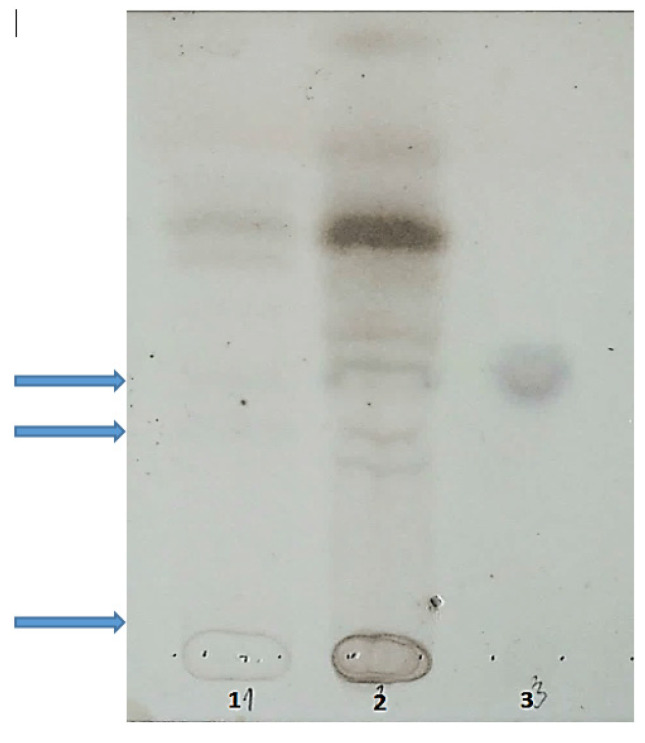
Thin-layer chromatography of sialooligosaccharides in urine. Path 1—healthy donor, path 2—described patient, path 3—free sialic acid. Desalted urine samples were mounted on the Silica G-60 precoated plate, developed in n-butyl alcohol:glacial acetic acid:water (50:25:25) and stained with resorcinol reagent. Arrows indicate sialooligosaccharides, which were slightly violet in the original image of the TLC plate. Bands of oligosaccharides in the control person were brown.

**Table 1 ijms-23-11424-t001:** Results of lysosomal enzymes activity and lysosphingolipids concentration in dried blood spot of the reported patient.

Lysosomal Enzyme/Lysosphingolipid	Enzyme Activity/Biomarker Concentration in Dried Blood Spot	Reference Values
**Lysosphingolipid concentration (ng/mL)**		
globotriaosylsphingosine (lyso-Gb3)	0.9	0.05–3.0
lysosphingomyelin (lyso-SM)	6.04	0.2–15
lysosphingomyelin-509 (lyso-SM-509)	2.9	0.15–3.7
hexosylsphingosines (galactosylsphingosine + glucosylsphingosine)	7.79	0.2–10
lyso-monosialoganglioside GM1 (lyso-GM1)	0	0–10
lyso-monosialoganglioside GM2 (lyso-GM2)	0	0–10
lysosulfatide	0	0–10
**Lysosomal enzymes activity (µmol/hr/L)**		
galactosylceramidase (GALC)	2.84	0.7–10
acid alpha-glucosidase (GAA)	5.12	1–25
alpha-galactosidase A (GLA)	5.53	0.8–15
beta-glucosidase (ABG)	4.12	1.5–25
acid sphingomyelinase (ASM)	5.68	1.5–25
alpha-L-iduronidase (IDUA)	4.86	1–25
N-Acetyl-Alpha-Glucosaminidase (NAGLU)	5.52	1–20
N-acetylgalactosamine 6-sulfatase (GALNS)	1.33	0.5–5
arylsulfatase B (ARSB)	4.10	1–15
beta-galactosidase (GLB1)	6.61	2–30
beta-glucuronidase (GUSB)	14.28	10–65
iduronate-2-sulfatase (ID2S)	18.5	10–50
tripeptidyl peptidase 1 (TPP1)	40.09	15–85

**Table 2 ijms-23-11424-t002:** Comparison between our patient and other MPS-PS described patients.

References	No. of Patients	Ethnicity	Consanguinity	Renal Phenotype	Hematological Phenotype	Cardiological Phenotype	Immunological Phenotype	Other	Follow-Up
This report	1	Caucasian	Probably yes	Slight proteinuria with normal kidney function	Normal results of hematological studies	Stable heart disease—mild mitral stenosis	Decreased serum IgG concentration	Recurrent peripheral edemas, autism spectrum disorder, visceral obesity, fetal ascites	Alive; last follow-up: 12 years
Gurinova et al., 2014 [8]	11	Yakut	Yes	Nephromegaly in 3/11	Not reported	Congenital heart defects in 7/11; heart failure and pulmonary hypertension	Not reported	Not reported	9/11 died till 2 y due to cardiorespiratory failure
Dursun et al., 2016 [9]	2	Turkish/ Yakut	Yes	Proteinuria in 2/2, renal biopsy—segmental/global sclerosis, periglomelular fibrosis	Anemia in 2/2	Not reported	Not reported	Not reported	2/2 died at the age of 6 years and 3 months (respiratory and renal failure) and 6 months (cardiopulmonary failure), respectively
Kondo et al., 2017 [10]	13	Yakut	Yes	Proteinuria in 13/13 while nephritic syndrome in 4/13; autopsy findings in 1 of them—significant grade of glomerular hyalinization, accumulation of lymphocytes in the renal interstitium	Anemia in 13/13, thrombocytopenia in 12/13, leukocytopenia in 8/13; bone marrow hypoplasia in 2/3	Congenital heart defects: patent ductus arteriosus in 7/13, atrial septal defect in 7/13; hypertrophic cardiomyopathy in 9/13	Not reported	Not reported	11/13 died of cardiorespiratory failure at approximately 1 to 2 y
Pavlova et al., 2019 [11]	5	Yakut	Yes	Nephrotic syndrome (full-blown) in 4/5	Anemia in 5/5, thrombocytopenia in 4/13, leukopenia in 4/5; coagulation defects with episodic intestinal bleeding in 2/5	Congenital heart defects: patent ductus arteriosus in 2/5	Low serum IgG concentration in 4/5	Not reported	
Vasilev et al., 2020 [12]	1	Yakut	No	Not reported	Anemia	Insufficiency of aortic valve, mitral and tricuspid valve regurgitation, pulmonary hypertension (1 y 9 months)	Not reported	Not reported	Death at 1 y 10 months due to respiratory insufficiency followed by multiple organ failure
Faraguna et al., 2022 [13]	1	Moroccan	Yes	Tubulopathy with low molecular weight proteinuria	Iron-refractory microcytic anemia, transient mild thrombo-cytopenia	Severe mitral insufficiency with atrial dilatation	Secondary hemophagocytic lymphohistiocytosis	Nonautoimmune subclinical hypothyroidism	Pneumonia complicated by respiratory insufficiency requiring orotracheal intubation at 2 y, development of a secondary hemophagocytic lymphohistiocytosis during septic shock due to pneumonia
Sofronova et al., 2022 [14]	5	Yakut	Yes	Proteinuria in 5/5, mild kidney damage (defined as eGFR < 90) in 4/5, elevated serum uric acid level in 4/5; autopsy findings—foamy podocytes, chronic interstitial inflammation, periglomelular fibrosis	Progressive anemia in 5/5, low within reference range platelets count in 5/5, bone marrow histology—hypocellular fatty marrow, absence of erythroblastic islands and megakaryocytes	Not reported	Not reported	Growth retardation in several patients; 5/5 patients had below-average weight	5/5 died—precise cause of death not known
Pavlova et al., preprint published on 30 August 2022 [15]	1	Southern European/Mediterranean	Not reported	Proteinuria in childhood	Normal results of hematological studies	Not reported	Normal serum IgG concentration	Autism spectrum disorder, intellectual disability	Alive early 20s

## Data Availability

All data generated or analysed during this study are included in this published article.

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
