# Peer review of "Mucopolysaccharidosis-Plus Syndrome: Report on a Polish Patient with a Novel VPS33A Variant with Comparison with Other Described Patients"

_ijms, 2022, doi:10.3390/ijms231911424_

Round 1

Reviewer 1 Report

Lipiński et al. present a case study for a patient with a novel molecular variant suggestive of an attenuated form of MPS-plus syndrome. While the case study is sound, I do feel there are some areas which need to be addressed.

1.       I would like to point the authors to a pre-print by Pavlova et al. which describes the same novel variant in an Israeli patient. Comparison to this should be made.

https://www.medrxiv.org/content/10.1101/2022.08.27.22279208v1

2.       I’m not sure that references 1-3 are the most appropriate for the introduction to MPS. There are many more comprehensive references that could be used. The book chapter by Neufeld and Muenzer in The Metabolic and Molecular Bases of Inherited Disease comes to mind.

3.       In the introduction, it would be beneficial to expand on the clinical and biochemical features of MPS/MPS-PS.

4.       Please expand on the methods presented in the paper, either by reference to published methods or details in the text or supplementary materials. As the methods currently stand, they could not be reproduced. Specifically:

-          Page 2, line 90. What buffer was used? What percent of aqueous NaCl? There is no mention of volumes. What temperature was incubation? What proportion was the mixture of water/acetonitrile and percentage of formic acid?

-          Page 3, line 95. Needs details on LC-MS/MS methods, including mobile phases and gradients, source conditions, method of quantification (MRM mode?), etc.

-          Page 3, line 100. The authors mention analysis of oligosaccharides and sialo-oligosaccharides. The cited papers use orcinol/sulphuric acid to visualise hexose-containing molecules, but there is no indication of how sialo-oligosaccharides were visualised.

5.       Please include the TLC images showing increased sialo-oligosaccharides. Patients with impaired renal function may have increased urine oligosaccharides but not in patterns typically seen in oligosaccharidoses patients. Could the increase in oligosaccharides seen here (and in the other MPS-PS patients) simply reflect this?

6.       Table 2 is almost impossible to read. While I appreciate this is a lot of data to try and show, this is one of the most important points of the manuscript and needs to be clear for the reader. Please include the patient from point 1, above.

7.       Page 11, line 238. Please consider altering “relatively mild” to attenuated in line with nomenclature used for other LSDs.

8.       Strengthening the correlation between clinical and biochemical results is needed. For example, GAG electrophoresis has been shown to be unreliable, particularly in attenuated forms of MPS but the link with the attenuated form of MPS-PS presented here has not been clearly made. Furthermore, HS has long been associated with neurological impairment in MPS. The presence of autistic features with the absence of HS accumulation adds weight to GAG accumulation being a consequence of disease and not a cause.

Author Response

Reviewer 1

We are very grateful to the Reviewer for his/her comments.

Lipiński et al. present a case study for a patient with a novel molecular variant suggestive of an attenuated form of MPS-plus syndrome. While the case study is sound, I do feel there are some areas which need to be addressed.

1.       I would like to point the authors to a pre-print by Pavlova et al. which describes the same novel variant in an Israeli patient. Comparison to this should be made.

https://www.medrxiv.org/content/10.1101/2022.08.27.22279208v1

The following sentence was added into the main text of manuscript: When this paper was being reviewed, a pre-print reporting a MPSPS patient with the homozygous c.599G>C variant of VPS33A appeared which corrorborated conclusions presented in our work.

  1. I’m not sure that references 1-3 are the most appropriate for the introduction to MPS. There are many more comprehensive references that could be used. The book chapter by Neufeld and Muenzer in The Metabolic and Molecular Bases of Inherited Disease comes to mind.

It was corrected as advised. Two more comprehensive references were added, as following:

Tomatsu S, Lavery C, Giugliani R, Harmatz P, Scarpa M, Węgrzyn G, Orii T (Eds.). Mucopolysaccharidoses Update (2 Volume Set). Metabolic Diseases - Laboratory and Clinical Research. New York: Nova Science Publishers; 2018.

Neufeld EF, Muenzer J. The Mucopolysaccharidoses. In: Valle DL, Antonarakis S, Ballabio A, Beaudet AL, Mitchell GA (Eds.). The Online Metabolic and Molecular Bases of Inherited Disease. McGraw Hill; 2019.

  1. In the introduction, it would be beneficial to expand on the clinical and biochemical features of MPS/MPS-PS.

It was corrected as advised.

The clinical features of MPS differ depending on the specifying enzyme deficiency and accumulated type of GAGs, however the most specific symptoms include a short stature (with the most severe growth impairment in MPS IVA), skeletal symptoms (dysostosis multiplex, joint stiffness but hypermobility and unique skeletal abnormalities in MPS IVA), neurodegeneration (progressive cognitive decline especially in MPS III but also MPS IH, neuronopathic MPS II), coarse facial features, cardiac (valvular) disease (MPS I, II, VI).

In 2017, a new disease assumed as Mucopolysaccharidosis-Plus Syndrome (MPS-PS) was included in the Online Mendelian Inheritance in Man® (OMIM®) database [7]. The name of the disease stems from the presence of only some clinical or biochemical features resembling MPS (including dysostosis multiplex in some cases, thickened facial features, valvular disease, psychomotor retardation, slightly increased urinary GAGs excretion), and additional signs and symptoms.

  1. Please expand on the methods presented in the paper, either by reference to published methods or details in the text or supplementary materials. As the methods currently stand, they could not be reproduced. Specifically:

    -          Page 2, line 90. What buffer was used? What percent of aqueous NaCl? There is no mention of volumes. What temperature was incubation? What proportion was the mixture of water/acetonitrile and percentage of formic acid?

    -          Page 3, line 95. Needs details on LC-MS/MS methods, including mobile phases and gradients, source conditions, method of quantification (MRM mode?), etc.

It was corrected as following:

The IS, substrates, and assay buffers of panel 1 (GALC, GAA, GLA, ABG, ASM and IDUA) and panel 2 (I2S, NAGLU, GALNS, ARSB, GLB1, GUSB and TPP1) were commercially purchased from PerkinElmer, Inc. (Waltham, MA, USA).

The protocol of the multiplex enzyme assay is described as following.  All of the assays were carried out with a 3.2 mm punch in 30 μL of assay cocktail in a polypropylene 96-well plate and incubated at 37 °C for 18 h. To terminate this enzyme reaction, a mixture of methanol/ethyl acetate (50/50, 100 μL) was added.  Solvent extraction was carried out by adding 400 µl of ethyl acetate and 300 µl of deionized water (for panel 1) and 400 µl of ethyl acetate and 300 µl of 0.5M NaCl (for panel 2), followed by centrifugation for 5 minutes at 4000 rpm. 300 µl of the organic layer was transferred to a new 96-well plate and dried under a stream of nitrogen at 400C. The dried precipitate was dissolved in 100 µl solvent mixture (acetonitrile: deionized water (80:20) containing 0.2% formic acid). The enzyme activity was calculated by the formula: A = ((P/IS) x СIS)/(3.1 x ti), where A - enzyme activity expressed as micromoles of product per litre of whole blood per hour (µM/l/h), P - product peak area produced during the enzymatic reaction, IS - peak area of internal standard, CIS - the concentration of the internal standard in μM, ti - incubation time, 3.1 - volume of sample in µl.

Samples were measured using an LC-30 Nexera System (Shimadzu Corporation, Kyoto, Japan) and a tandem mass spectrometer QTrap 4500 (ABSciex, USA) equipped with a positive electrospray ionization by selected ion monitoring mode (Multiple Reaction Monitoring, MRM).

Detailed characteristics is presented below (not included in the manuscript).

MS instrument parameters for panel 1.

Curtain Gas (CUR)

30 L/h

Collision Gas (CAD)

Medium

Ion Spray Voltage

5500 V

Temperature (TEM)

4800C

Ion Sourse Gas 1 (GS1)

50 L/h

Ion Sourse Gas 1 (GS2)

65 L/h

Dwell time

18 ms

Declustering potential (DP)

40 V

Entrance Potential (EP)

10 V

Collision Cell Exit potential (CXP)

10 V

MRM parameters for panel 1.

Precursor ion

(m/z)

Product ion

(m/z)

CE

(V)

GALC-P

412.20

264.20

30

GALC-IS

417.20

264.20

30

GAA-P

498.20

398.20

32

GAA-IS

503.20

403.20

32

GLA-P

484.20

384.20

32

GLA-IS

489.20

389.20

32

ABG-P

384.20

264.20

28

ABG-IS

391.20

271.20

28

ASM-P

398.20

264.20

30

ASM-IS

405.20

264.20

30

IDUA-P

426.20

317.20

25

IDUA-IS

431.20

322.20

25

P, product of enzyme reaction

IS, internal standard.

DP, declastering potencial

CE, collision energy

HPLC method for panel 1.

Column

XBridge BEH C18 (Waters, USA).

Particle diameter: 3.5 mm

Internal diameter: 2.1 mm

Length: 50 mm

Column temperature

60°C

Wash solvent

80% Methanol in water

Mobile phase A

0.2% formic acid in water

Mobile phase B

0.2% formic acid in acetonitrile

Gradient (% B)

0 - 0.50 min: 10% B – 98% B

0.50 - 2.00 min: 98% B

2.00 – 3.00 min: 10% B

Flow rate

0.4 mL/min

Injection volume

10 mL

Autosampler temperature

15°C

MS instrument parameters for panel 2.

Curtain Gas (CUR)

20 L/h

Collision Gas (CAD)

Medium

Ion Spray Voltage

5500 V

Temperature (TEM)

6000C

Ion Sourse Gas 1 (GS1)

50 L/h

Ion Sourse Gas 1 (GS2)

65 L/h

Dwell time

20 ms

Entrance Potential (EP)

10 V

Collision Cell Exit potential (CXP)

10 V

MRM parameters for panel 2.

Precursor ion

(m/z)

Product ion

(m/z)

DP

(V)

CE

(V)

NAGLU-P

420.30

311.30

65

21

NAGLU-IS

423.30

314.30

65

21

GALNS-P

685.40

373.30

80

35

GALNS-IS

690.40

378.30

80

35

ARSB-P

657.40

345.20

80

33

ARSB-IS

662.40

350.30

80

33

GLB1-P

436.30

336.30

50

21

GLB1-IS

439.30

339.30

50

21

GUSB-P

434.30

325.30

80

34

GUSB-IS

439.30

330.30

80

34

I2S-P

644.40

359.30

80

30

I2S-IS

649.30

364.30

80

30

TPP1-P

350.30

250.30

36

20

TPP1-IS

359.30

251.30

36

20

P, product of enzyme reaction

IS, internal standard.

DP, declastering potencial

CE, collision energy

HPLC method for panel 2.

Column

Fusion-RP (Phenomenex, USA).

Particle diameter: 4.0 mm

Internal diameter: 2.1 mm

Length: 50 mm

Column temperature

60°C

Wash solvent

80% Methanol in water

Mobile phase A

0.2% formic acid in water

Mobile phase B

0.2% formic acid in acetonitrile

Gradient (% B)

0 – 2.00 min: 10% B – 80% B

2.00 - 2.50 min: 80% B

2.50 – 3.50 min: 10% B

Flow rate

0.5 mL/min

Injection volume

5 mL

Autosampler temperature

15°C

-          Page 3, line 100. The authors mention analysis of oligosaccharides and sialo-oligosaccharides. The cited papers use orcinol/sulphuric acid to visualise hexose-containing molecules, but there is no indication of how sialo-oligosaccharides were visualised.

  1. Please include the TLC images showing increased sialo-oligosaccharides. Patients with impaired renal function may have increased urine oligosaccharides but not in patterns typically seen in oligosaccharidoses patients. Could the increase in oligosaccharides seen here (and in the other MPS-PS patients) simply reflect this?

    It was corrected as advised.

Thin-layer chromatography (TLC) of oligosaccharides and sialooligosaccharides was performed according to the methods described by Blom et al., Humbel and Collart, and Svennerholm with own modification, which was the introduction of a preliminary step of desalting the urine samples on mini-columns with anionite AG 1X8, 200-400 mesh (Bio Rad) and cationite Dowex, 50X8-200, 100-200 mesh (Sigma-Aldrich) [16-19].

Figure 5. Thin-layer chromatography of sialooligosaccharides in urine.
Path 1 – control person, path 2 – described patient, path 3 – free sialic acid;

Desalted urine samples were mounted on the Silica G-60 precoted plate, developed in n-butyl alcohol: glacial acetic acid: water (50: 25: 25) and stained with resorcinol reagent.

Arrows indicate sialooligosaccharides, which were slightly violet in original image of the TLC plate. Bands of oligosaccharides in control person were brown.

  1. Table 2 is almost impossible to read. While I appreciate this is a lot of data to try and show, this is one of the most important points of the manuscript and needs to be clear for the reader. Please include the patient from point 1, above.

The style of Table 2 was corrected to be clear and easy to follow. The patient from pre-print was added.

References

No. of patients

Ethnicity

Consanguinity

Renal phenotype

Hematological phenotype

Cardiological phenotype

Immunological phenotype

Other

Follow-up

This report

1

Caucasian

Probably yes

Slight proteinuria with normal kidney function

Normal results of hematological studies

Stable heart disease – mild mitral stenosis

Decreased serum IgG concentration

Recurrent peripheral edemas, autism spectrum disorder, visceral obesity, foetal ascites

Alive; last follow-up: 12 years

Gurinova et al., 2014

11

Yakut

yes

Nephromegaly in 3/11

Not reported

Congenital heart defects in 7/11; heart failure and pulmonary hypertension

Not reported

Not reported

9/11 died till 2 y due to cardiorespiratory failure

Dursun et al., 2016

2

Turkish/

Yakut

yes

Proteinuria in 2/2, renal biopsy – segmental/global sclerosis, periglomelular fibrosis

Anemia in 2/2

Not reported

Not reported

Not reported

2/2 died at the age of 6 years and 3 months (respiratory and renal failure) and 6 months (cardiopulmonary failure), respectively

Kondo et al., 2017

13

Yakut

yes

Proteinuria in 13/13 while nephritic syndrome in 4/13; autopsy findings in 1 of them - significant grade of glomerular hyalinization, accumulation of lymphocytes in the renal interstitium

Anemia in 13/13, thrombocytopenia in 12/13, leukocytopenia in 8/13; bone marrow hypoplasia in 2/3

Congenital heart defects: patent ductus arteriosus in 7/13, atrial septal defect in 7/13; hypertrophic cardiomyopathy in 9/13

Not reported

Not reported

11/13 died of cardiorespiratory failure at approximately 1 to 2 y

Pavlova et al., 2019

5

Yakut

Yes

Nephrotic syndrome (full-blown) in 4/5

Anemia in 5/5, thrombocytopenia in 4/13, leukopenia in 4/5; coagulation defects with episodic intestinal bleeding in 2/5

Congenital heart defects: patent ductus arteriosus in 2/5

Low serum IgG concentration in 4/5

Not reported

Vasilev et al., 2020

1

Yakut

No

Not reported

Anemia

insufficiency of aortic valve, mitral and tricuspid valve regurgitation, pulmonary hypertension [1 y 9 mo]

Not reported

Not reported

Death at 1 y 10 mo due to respira-tory insuffi-ciency followed by multi-ple organ failure

Faraguna et al., 2022

1

Moroccan

Yes

Tubulopathy with low molecular weight proteinuria

Iron-refractory microcytic anemia, transient mild thrombo-cytopenia

severe mitral insufficiency with atrial dilatation

Secondary hemophagocytic lymphohistiocytosis

Non-autoimmune subclinical hypothyroidism

Pneumonia complicated by respiratory insufficiency requiring orotracheal intubation at 2 y, development of a secondary hemophagocytic lymphohistiocytosis during septic shock due to pneumonia

Sofronova et al., 2022

5

Yakut

Yes

Proteinuria in 5/5, mild kidney damage (defined as eGFR < 90) in 4/5, elevated serum uric acid level in 4/5; autopsy findings – foamy podocytes, chronic interstitial inflammation, periglomelular fibrosis

Progressive anemia in 5/5, low within reference range platelets count in 5/5, bone marrow histology – hypocellular fatty marrow, absence of erythroblastic islands and megakaryocytes

Not reported

Not reported

Growth retardation in several patients; 5/5 patients had below-average weight

5/5 died – precise cause of death not known

Pavlova et al., pre-print published on 30 Aug 2022

1

Caucasian

Not reported

Proteinuria in childhood

Normal results of hematological studies

Not reported

Normal serum IgG concentration

Autism spectrum disorder, intellectual disability

Alive early 20s

  1. Page 11, line 238. Please consider altering “relatively mild” to attenuated in line with nomenclature used for other LSDs.

It was corrected as advised.

  1. Strengthening the correlation between clinical and biochemical results is needed. For example, GAG electrophoresis has been shown to be unreliable, particularly in attenuated forms of MPS but the link with the attenuated form of MPS-PS presented here has not been clearly made.

Furthermore, HS has long been associated with neurological impairment in MPS. The presence of autistic features with the absence of HS accumulation adds weight to GAG accumulation being a consequence of disease and not a cause.

It was corrected as advised.

We would like to highlight that there is no correlation between clinical and biochemical (including urinary GAGs excretion) features of MPS-PS. Furthermore, HS has long been associated with neurological impairment in MPS. The presence of familial autistic features with the absence of HS accumulation adds weight to GAGs accumulation being a con-sequence of disease and not a cause.

Reviewer 2 Report

In the present work, authors describe the first Polish patient affected by Mucopolysaccharidosis-Plus syndrome, with a novel homozygous c.599G>C (p.Arg200Pro) VPS33A variant. The disease has been described only in 2017 and so far, only 38 patients have been identified. All the other patients display the same mutation in the VPS33A gene, encoding for the vacuolar protein sorting-associated protein 33A. They also share some clinical and biochemical features (including increased urinary GAG excretion) resembling MPS, and additional signs and symptoms.

Authors report 12 years follow-up, on this patient who shows some novel clinical features, including fetal ascites (resolved spontaneously), recurrent joint effusion and peripheral aedemas, normal growth and visceral obesity.

The paper is well written, however, I have some concerns summarized in the specific comments below.

Specific comments :

Figure 2. Please improve the legend of the figure, explaining the graphs.

Line 158. Authors refer to the patient as “the girl”, but without mentioning the sex before, making the presentation of results difficult to follow. Please use a unique term, like patient or subject.

Figure 4. The figure needs to be modified, as it has pen marks covering the bands. I would recommend to provide a better picture, without pen marks. Also the legend needs to be improved (part 3: control person, would be better healthy donor, for example).

Line 207 Please explain better the decision to investigate the VPS33A gene. Did authors observed similarities with reported cases of Mucopolysaccharidosis-Plus syndrome?

Table 2. Please improve the table, it is really hard to read the content of each column. Also the content presents a few typing mistakes.

Line 259 As authors report” Growth retardation as well as poor weight gain were also observed in several patients reported by Sofronova et al. [14]. Contrary to that, our patient had normal growth and was overweight (visceral obesity).” Is there any hypothesis that could explain this discrepancy?

Taking into consideration these comments I would recommend these minor revisions to the paper.

Author Response

Reviewer 2

Comments and Suggestions for Authors
In this manuscript, the authors report the first case of a Polish patient affected by the Mucopolysaccharidosis-Plus Syndrome (MPS-PS). In contrast to the other 38 patients described to date which exhibit the c.1492C>T (p.Arg498Trp) variant of the VPS33A gene, a novel homozygous c.599G>C (p.Arg200Pro) VPS33A variant  was detected in the Polish patient. The authors provide a detailed analysis of both biochemical and clinical data recorded for this patient as well as a comparison with the phenotype of the other MPS-PS described cases. The manuscript is well written, the methodology rigorous and appropriate, and the topic is interesting. The new VPS33A mutation found in the Polish patient provides more information on a disease whose physiopathology has yet to be established. 

Minor points:

-       Pag. 2, ln.58: spell “GAGs”

-       Please, define the abbreviations at the first time you use them [i.e., page 2, ln. 85 “Dry Blood Spot (DBS)”, ln.86 “Internal Standard (IS)”, ln. 91 “Milli-Q water (MQ)”, ln. 94 “liquid chromatography-tandem mass spectrometry (LC-MS/MS)”, page 3 “ln. 103, thin-layer chromatography (TLC)”, ln. 112 “whole exome sequencing (WES)”, page 4, ln. 155 “percentile (pc)”, page 5, ln. 176 “total body water (TBW)”],

-       Page 2, ln. 86: spell “substrates”

-       Page 3, lns 108-109: Why do not include the methodology of “bioelectric impedance analysis” here, instead than as supplementary material?

-       Page 11, ln. 266: spell “KS” instead of “keratan sulfate”

-       Page 6, ln. 189; Page 11, ln. 270: spell CS” instead of “chondroitin sulfate”

We are very grateful to the Reviewer for his/her comments.

All of the aforementioned issues were corrected as advised. The methodology of BIA was included in the main text of manuscript.

Reviewer 3 Report

In this manuscript, the authors report the first case of a Polish patient affected by the Mucopolysaccharidosis-Plus Syndrome (MPS-PS). In contrast to the other 38 patients described to date which exhibit the c.1492C>T (p.Arg498Trp) variant of the VPS33A gene, a novel homozygous c.599G>C (p.Arg200Pro) VPS33A variant  was detected in the Polish patient. The authors provide a detailed analysis of both biochemical and clinical data recorded for this patient as well as a comparison with the phenotype of the other MPS-PS described cases. The manuscript is well written, the methodology rigorous and appropriate, and the topic is interesting. The new VPS33A mutation found in the Polish patient provides more information on a disease whose physiopathology has yet to be established.  

Minor points:

-       Pag. 2, ln.58: spell “GAGs”

-       Please, define the abbreviations at the first time you use them [i.e., page 2, ln. 85 “Dry Blood Spot (DBS)”, ln.86 “Internal Standard (IS)”, ln. 91 “Milli-Q water (MQ)”, ln. 94 “liquid chromatography-tandem mass spectrometry (LC-MS/MS)”, page 3 “ln. 103, thin-layer chromatography (TLC)”, ln. 112 “whole exome sequencing (WES)”, page 4, ln. 155 “percentile (pc)”, page 5, ln. 176 “total body water (TBW)”],

-       Page 2, ln. 86: spell “substrates”

-       Page 3, lns 108-109: Why do not include the methodology of “bioelectric impedance analysis” here, instead than as supplementary material?

-       Page 11, ln. 266: spell “KS” instead of “keratan sulfate”

-       Page 6, ln. 189; Page 11, ln. 270: spell CS” instead of “chondroitin sulfate”

Author Response

Reviewer 3

In the present work, authors describe the first Polish patient affected by Mucopolysaccharidosis-Plus syndrome, with a novel homozygous c.599G>C (p.Arg200Pro) VPS33A variant. The disease has been described only in 2017 and so far, only 38 patients have been identified. All the other patients display the same mutation in the VPS33A gene, encoding for the vacuolar protein sorting-associated protein 33A. They also share some clinical and biochemical features (including increased urinary GAG excretion) resembling MPS, and additional signs and symptoms.

Authors report 12 years follow-up, on this patient who shows some novel clinical features, including fetal ascites (resolved spontaneously), recurrent joint effusion and peripheral aedemas, normal growth and visceral obesity.

The paper is well written, however, I have some concerns summarized in the specific comments below.

We are very grateful to the Reviewer for his/her comments.

Specific comments :

Figure 2. Please improve the legend of the figure, explaining the graphs.

Both, figure 2 and its legend, were corrected.

Line 158. Authors refer to the patient as “the girl”, but without mentioning the sex before, making the presentation of results difficult to follow. Please use a unique term, like patient or subject.

It was corrected as advised.

Figure 4. The figure needs to be modified, as it has pen marks covering the bands. I would recommend to provide a better picture, without pen marks. Also the legend needs to be improved (part 3: control person, would be better healthy donor, for example).

Unfortunately, we could not provide the fure 4 without pen marks covering the bands.

Regarding figure 3, it was corrected as advised.

Line 207 Please explain better the decision to investigate the VPS33A gene. Did authors observed similarities with reported cases of Mucopolysaccharidosis-Plus syndrome?

Due to an unknown etiology of presented features, whole exome sequencing (WES) was commenced. Based on the WES data the homozygous variant in VPS33A gene (NM_022916.6:c.599G>C) have been pointed for further verification.

Table 2. Please improve the table, it is really hard to read the content of each column. Also the content presents a few typing mistakes.

Table 2 was improved.

Line 259 As authors report” Growth retardation as well as poor weight gain were also observed in several patients reported by Sofronova et al. [14]. Contrary to that, our patient had normal growth and was overweight (visceral obesity).” Is there any hypothesis that could explain this discrepancy?

We provide a report of MPS-PS patient with an attenuated phenotype. Regarding overweight, it was observed as a familial feature.

Taking into consideration these comments I would recommend these minor revisions to the paper.

Round 2

Reviewer 1 Report

Thanks to the authors for their hard work in addressing the concerns that I had. There are just a couple of points and some typos I would like to add.

1.       It would be helpful to place the LC-MS/MS conditions supplied in the reviewer rebuttal in supplementary materials and refer to this on page 3, line 115 if these are not published elsewhere.

2.       Abstract, line 23 – this should now be 39 with the addition of the new patient, which should be mentioned here.

3.       Page 3, line 105 – change to 40oC.

4.       Page 5, Figure 1 legend. Key repeated for I, II, III.

5.       Page 7, Figure 4 legend – cellulose.

6.       Page 8, Figure 5 legend – precoated.

7.       Page 12, line 290 – corroborated.

8.       Note, to editor, Page 13, lines 321 and 325. Words deleted on marked up version not removed here.

Author Response

We are very grateful to the Reviewer for his/her comments.   The manuscript was corrected as advised. LC-MS/MS conditions were described in Supplementary Material.